# SiC ultra -thin films for high-performance quantum spacecraft applications fabricated via nanosecond pulsed laser deposition and doping

Praveen Kumar Kanti[1], Prashantha Kumar H.G.[2]*, V. Vicki Wanatasanappan[1]*, Abhinav Kumar[3,4,5]*, Dhouha Choukaier[6], Subbulakshmi Ganesan[7], Tekalign Negash Kebede [8]*

1 Institute of Power Engineering, Universiti Tenaga Nasional, Jalan IKRAM-UNITEN, Selangor, Malaysia, 2 Department of aerospace engineering, School of Engineering (SoE), Dayananda Sagar University, Bangalore, India, 3 Department of Nuclear and Renewable Energy, Ural Federal University Named after the First President of Russia Boris Yeltsin, Ekaterinburg, Russia, 4 Centre for Research Impact & Outcome, Chitkara University Institute of Engineering and Technology, Chitkara University, Rajpura, Punjab, India, 5 Department of Mechanical Engineering and Renewable Energy, Technical Engineering College, The Islamic University, Najaf, Iraq, 6 Department of Basic Health Sciences, College of Foundation Year for Health Colleges, Princess Nourah bint Abdulrahman University, Riyadh, Saudi Arabia, 7 Department of Chemistry and Biochemistry, School of Sciences, JAIN (Deemed to be University), Bangalore, Karnataka, India, 8 Department of Accounting and Finance, College of Business and Economics, Hawassa University, Hawassa, Ethiopia

* prashanthakumar.hg@gmail.com (PKHG); vignesh@uniten.edu.my (VVW); drabhinav@iee.org (AK); tekan@hu.edu.et (TNK)

## Abstract

This research investigates the deposition, characterization of SiC ultra-thin films deposited by nanosecond pulsed $Nd^{3+}$ laser deposition technique and laser assisted doping. These SiC films possess better qualities in terms of surface roughness varying from 2–5 nm. Using an atom probe tomography (APT) and a transmission electron microscope (TEM), key elemental maps showed desirable concentrations of Si of ~50 at.% and C of ~48 at.% with minor elemental contents including O, Al, N, and B. These distributions indicate the material's structural and chemical performance under laser assisted doping. I-V measurements reveal better electronic response, including low turn-on voltage, and high carrier mobility that makes these films suitable for quantum spacecrafts. This research stresses the review of laser fluence and doping as important parameters for modifying SiC thin films for future electronics and space applications.

## 1. Introduction

Silicon carbide (SiC), a wide bandgap (WBG) semiconductor with a bandgap of 3.2 eV, has become a cornerstone material for advanced electronics and space technology. Its exceptional properties, including high electron mobility (900 cm²/V·s), hole mobility (115 cm²/V·s), critical electric field (3 MV/cm), and thermal conductivity (4.9 W/cm·K),

**Data availability statement:** All data is available within the manuscript and the supporting information files.

**Funding:** Princess Nourah bint Abdulrahman University Researchers Supporting Project number (PNURSP2025R855), Princess Nourah bint Abdulrahman University, Riyadh, Saudi Arabia.

**Competing interests:** The authors have declared that no competing interests exist.

significantly surpass those of silicon (WBG ~ 1.2 eV), enabling the development of high-temperature and high-power electronic devices [1–3]. The capability of SiC to sustain higher blocking voltages and reduced leakage currents at elevated temperatures makes it particularly attractive for quantum systems in spacecraft, where reliability under extreme conditions is paramount [4,5]. Effective thermal management is critical, as quantum devices, like superconducting qubits, demand cryogenic temperatures to function. Radiation hardening is another vital aspect, ensuring that quantum systems remain functional and maintain coherence in the presence of high-energy cosmic rays and space radiation. Wide bandgap materials, such as SiC, are increasingly employed for radiation-resistant power electronics in these demanding conditions.

Despite these advantages, the fabrication of single-crystal SiC thin films for quantum applications poses challenges. SiC exists in over 250 polytypes, with cubic (3C-SiC) and hexagonal (4H-SiC and 6H-SiC) configurations being most relevant for electronic applications [6]. Reaction-bonded SiC (RB-SiC), manufactured via molten silicon infiltration at ~1450°C, is a cost-effective material with potential for power electronics. However, achieving defect-free, high-purity SiC thin films for quantum device applications remains an unmet need [7]. To address these challenges, pulsed laser deposition (PLD) has emerged as a versatile technique for the fabrication of SiC thin films. PLD enables the deposition of both crystalline and amorphous films at low temperatures (<800°C) while maintaining precise stoichiometric control. Employing lasers such as ArF (193 nm), KrF (248 nm), and $Nd^{3+}$(355 nm), PLD facilitates multilayer deposition without significant contamination [8–10]. However, optimizing the doping of SiC, particularly with low-diffusion-coefficient impurities, requires advanced methods. Selective pulsed laser-assisted doping (LAD) is a promising solution, enabling controlled impurity incorporation and minimizing secondary phase formation [11–13]. For advanced quantum systems, the characterization of doped SiC thin films at an atomic scale is critical. Atom probe tomography (APT) has emerged as an advanced analytical technique for three-dimensional mapping of dopant positions and compositional analysis within nanometric volumes. APT provides unmatched sensitivity for detecting both light and heavy elements, overcoming limitations of conventional methods [14–17].

With these advantages and critical future need, the current study investigates the synthesis and characterization of SiC ultra-thin films for quantum spacecraft applications. RB-SiC thin films were fabricated using PLD and enhanced through selective LAD to incorporate suitable impurities. The Pulsed Laser Deposition (PLD) method provides many benefits when used for thin film deposition in comparison to CVD and sputtering. The technique delivers specific stoichiometric measurements which leads to producing highly refined SiC films while reducing their impurity content. PLD enables substrate-friendly low-temperature depositions that can happen below 800°C. The process enables both high crystallinity and selective doping which enhances material characteristics needed for quantum and electronic device systems [18]. The film quality achieved and scalability together with reduced contamination are better in PLD systems when compared to traditional CVD and sputtering technological approaches. Further, PLD operations spans from 12 to 15 hours based on

the laser parameters. The processing duration for CVD extends beyond 12–15 hours coupled with elevated temperatures higher than 1400°C because PLD operates with superior efficiency [19]. The precise doping capability offered by PLD stands vital for the fabrication of space electronics because it enables necessary electrical property tuning. Its operation takes place at reduced temperatures thus enabling its use with heat-sensitive space application components. PLD represents an excellent production technique when fabricating SiC films for space systems exposed. Advanced analytical techniques, including APT chemical reconstruction, nanoindentation, and surface profiling, were employed to evaluate the films' microstructural and mechanical properties. Furthermore, thermal probe I-V characterizations were performed to assess the electronic behaviour of these films. The findings of this study provide critical insights into the development of SiC thin films for powering quantum spacecraft, bridging the gap between material innovation and space technology. Such developed WBG devises tested in space to understand its performance under microgravity and radiations to qualify for space missions

## 2. Materials and methods

Pulsed laser deposition (PLD) was performed using a Q-switched Nd-YAG laser (Quantel, BRILB IR-10) with a wavelength of 355 nm, operating at a frequency of 10 Hz and a pulse width of 4 ns. The deposition utilized a reaction-bonded silicon carbide (RB-SiC) target supplied by Carborundum Universal Limited (CUMI) and a silicon (Si [100]) substrate under a constant supply of argon (Ar) gas (Fig 1). Additionally, Liquid Immersed Laser-Assisted Doping (LAD) was carried out at room temperature using a Q-switched Nd-YAG laser (Quantel Q-smart-850) with the same wavelength (355 nm), frequency (10 Hz), and pulse width (4 ns) as shown in Fig 1b. The detailed process experimental conditions are summarized in Table 1. The surface profile and nanoindentation hardness of the thin films were measured after each experiment using a non-contact profilometer (Bruker-3D) and a nano indenter (Hysitron Ltd, TI 750), respectively. Current-voltage (I-V) characteristics were evaluated using a four-probe station (Everbeing International Corporation, Taiwan).

The doped RB-SiC thin films were characterized at the atomic scale and their three-dimensional (3D) elemental distribution analyzed using atom probe tomography (APT) with a local electrode atom probe (LEAP™ 4000X HR, CAMECA Instruments). The region in RB-SiC thin as the selected region of interest for chemical composition reconstruction. APT

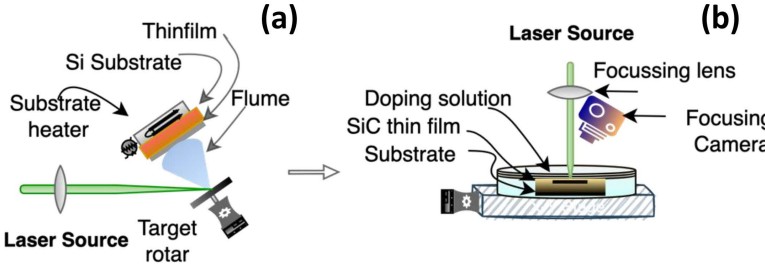

**Fig 1. Schematics of laser assist SiC thin film (a) deposition and (b) doping.**

**Table 1. Process parameters used for PLD & LAD.**

| Process | Parameters | Specifications |
| --- | --- | --- |
| PLD | Laser fluence (J/cm$^{-2}$) | 7 |
| | Substrate temperature (°C) | 800 |
| LAD | Laser fluence (J/cm$^2$) | 1.8 |
| | Doping & Source | Aluminum & AlCl$_3$ |

specimens were prepared using a site-specific lift-out method performed in a dual-beam scanning electron microscope equipped with a focused ion beam (SEM-FIB) utilizing gallium (Ga+) sources. To protect the selected site from damage during subsequent ion milling, a 1 μm thick protective layer of platinum was deposited. FIB milling was then employed to shape the specimen into a needle-like geometry, approximately 20 μm in length. Real-time monitoring during SEM imaging ensured precise site preparation. The resulting specimen, shaped as a $20 \times 4 \times 1$ μm³ wedge, was lifted out and mounted on pre-fabricated silicon (Si) posts for further processing. The annular milling process was continued to refine the wedge-shaped sample into the required needle geometry. These needle-shaped specimens were mounted, inserted into an ultra-high vacuum system, and cooled to cryogenic temperatures between 60 and 100 K. APT analysis was conducted by applying a positive voltage combined with nanosecond laser pulses at 30 pJ energy and a 100 kHz repetition rate. This facilitated layer-by-layer field evaporation of the material, where ions generated at each pulse impact were collected and their positional and spectral data were reconstructed for detailed chemical composition analysis.

The electron and hole mobility characteristics in SiC thin films are determined by Hall Effect experiments at different electric fields. The geometry of SiC samples identified and then applying ohmic contacts to their surface using nickel materials. SiC samples with ohmic contact placed on a Hall measurement system Ecopia HMS made of a current source, a magnetic field generator and the voltmeter. An adjustable same and fixed perpendicular magnetic field is used and a known current is passed through the SiC. Further, I-V characteristics were carried out at p-n junction at various temperature to understand the stability and its performance at extreme conditions.

## 3. Result and discussions

### 3.1. Microstructure and XRD analysis of SiC thin films

The microstructure analysis of the reaction-bonded SiC (RB-SiC) thin films, as shown in Fig 2(a) and (b), provides significant insights into the morphology, uniformity, and interface quality of the deposited thin films. Fig 2(a) highlights the SiC matrix with the presence of boron carbide ($B_4C$) particles, indicating the compositional inhomogeneity inherent in reaction-bonded SiC. These $B_4C$ particles act as residual phases, which can influence the mechanical and thermal properties of the SiC matrix. The distribution of these phases suggests that the molten silicon infiltration process has achieved partial densification, leaving behind some unreacted phases. This is consistent with previously reported challenges in RB-SiC processing, where achieving full densification without residual phases remains difficult. Fig 2(b) demonstrates the cross-sectional view of the SiC thin film deposited on the silicon substrate, revealing a uniform film thickness of approximately 400 nm. The interface between the SiC film and the silicon substrate appears smooth and defect-free, indicating good adhesion and minimal thermal stresses during deposition. Such uniformity is critical for applications in MEMS and quantum devices, where surface roughness and interface defects can significantly impact device performance [20,7].

The X-ray diffraction (XRD) analysis shown in Fig 2(c) presents the diffraction patterns for both 4H-SiC and 6H-SiC, which are two prominent polytypes of silicon carbide (SiC). These polytypes differ in their stacking sequences and exhibit distinct diffraction peaks due to their unique crystallographic structures. The diffraction patterns confirm the successful deposition of SiC thin films on the silicon substrate and the formation of both 4H-SiC and 6H-SiC phases. In the diffraction pattern, the characteristic peaks for 4H-SiC are observed at 2θ values of ~35°, ~60°, and ~72°, corresponding to reflections from (002), (112), and (114) planes, respectively [21,22]. Similarly, for 6H-SiC, peaks are observed at ~34°, ~58°, and ~70°, indicating reflections from (100), (103), and (006) planes [23]. substrate provides an epitaxial platform for SiC growth, as evidenced by the uniformity in peak intensity and the absence of spurious peaks from secondary phases.

The sharpness and intensity of these peaks indicate high crystallinity, suggesting minimal structural defects in the deposited thin films. The presence of distinct peaks for both polytypes highlights the coexistence of 4H-SiC and 6H-SiC in the deposited thin films, which is consistent with the typical behaviour of reaction-bonded SiC under specific deposition conditions. The observed variation in peak intensities between 4H-SiC and 6H-SiC suggests differences in their relative phase distributions and preferred orientations within the film. For instance, the dominance of the (002) peak in 4H-SiC

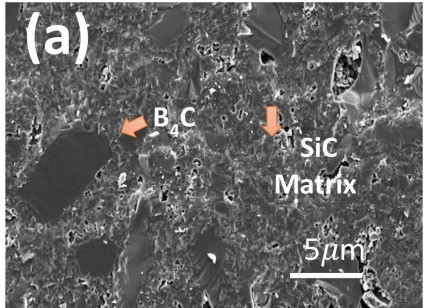

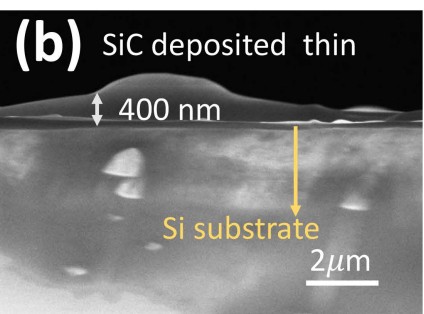

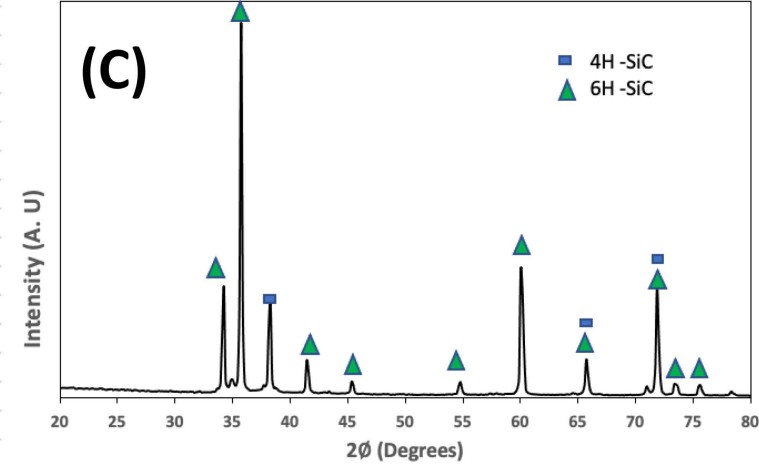

**Fig 2. Scanning Electron Microscopic (SEM) fractography of reaction bonded SiC indicates (a) SiC primary matrix with B$_4$C as additives (b) Transmission Electron Microscopic (TEM) of deposited thin film and corresponding (c) X-ray diffraction analysis (XRD).**

indicates a preferential c-axis orientation, which is advantageous for applications requiring high electron mobility and thermal conductivity [7,5]. Furthermore, the XRD results demonstrate the effectiveness of the pulsed laser deposition (PLD) technique in producing highly crystalline SiC thin films. The integration of a silicon

### 3.2. Thin-film composition reconstruction using atom probe tomography

Morphological and structural properties of lamellar specimen of SPS-SiC and RB-SiC thin films were characterized by high resolution transmission electron microscopy (TEM). The lamellar specimens were developed using the 'Forschungszentrum Karlsruhe' FEI Helios G4 UX dual beam Focused Ion Beam (FIB) system and Zeiss NVision SEM system. These tools allowed for site-specific sample preparation on the deposited thin films using a gallium (Ga$^+$) ion beam of 30 kV [14,24]. To prevent ion damage in the selected regions during high energy irradiation, a protective layer of platinum (Pt) was deposited over the regions of interest. The areas of coating was about 15μm long, 2.5μm wide and 1μm thick to act as a protective layer during milling. In the vicinity of these Pt-coated regions, Ga$^+$ ions were used to mill the specimens and these samples were later lifted off using an omni probe. The prepared lamellae were subsequently welded onto the TEM grid with Pt metal which provides a strong support [25,26].

The lamellae were finally thin to about 150–200 nm thickness and was done in the low energy Ga$^+$ sources in order to reduce the damages and the amorphization [12,27]. Specimen preparation of SiC thin films was carried out and the specimens were immediately examined using Tecnai F30 TEM instrument so as to prevent oxidation of the specimens.

Analyzing the thin films through high angular bright field scanning TEM, the authors established that the coatings were uniform in thickness and structure along their entire length on silicon (Si)substrate [10]. The TEM images presented in Fig 5 also give the total deposition thickness of SPS-SiC as 558.9 nm (Fig 3a) which is in agreement with the cross sectional image of RB-SiC as 593.62 nm (Fig 3b). These results also underline the uniform deposition, which was reached by using the applied technique the PLD method. Further, there was no signs of damages or amorphization observed in the lamellae during the cross-sectional TEM which approves the protective measure taken during the FIB milling and preparation of TEM samples [9]. Such characterization proves that the deposition process yielded high quality thin films with consistently thin layer which is suitable for power electronic device and micro electrical mechanical system applications. By incorporating Pt protection layers, the structural damage to the films is minimized, allowing correct characterization of their properties low energy milling [28,29].

The elemental concentration which is the second dimension represented across the analysed volume can also be obviously seen in the compositional depth profile (right). The guard composition of silicon and carbon—the major constituents of SiC—remains stable with atomic concentrations of ~ 50% and ~48%, respectively. The evidence of the Si and C distribution uniformity shows the high stoichiometric quality of the deposited thin films as the major quality factor that defines their mechanical and electrical properties in applications involving high power density and temperature [14,30]. Each spectrum has a very weak oxygen signal (O~0.43 at.%) which can be attributed to surface oxidation or contamination during the deposition or preparation process. Further, minor interstitial elements like aluminum (Al), nitrogen (N), vanadium (V), phosphorus (P), and boron (B) are observed, of which boron is characteristic of the reaction-bonded (RB-SiC) thin films only. It is possible that these are inherent with the target material or process environment, and their deliberate introduction for potential modulation of the electronic properties of the thin films [12,25].

The actual three dimensional atom map of the reconstructed form is shown in Fig 4 (left) which provides substantive evidence on the location of dopants and impurities. A constant distribution of Si and C throughout the investigated volume is related to the thin-film nature of the investigated samples. However, local dopant or impurity concentrations may indicate differences in a few process parameters or that the RB-SiC target material is intrinsically inhomogeneous. These clusters, though minor, could alter electrical and mechanical characteristic of the material in the deposition process if not well managed [31,16]. The characterization of the APT also shows the capability of achieving high sensitivity for light elements such as C, B and N, which are superior to methods like SIMS (Secondary Ion Mass Spectrometry) or EDS. In addition, the high spatial resolution, which lends APT the capability to map dopants and impurities at the atomic level, offers clues on how they influence the performance of the SiC thin film. Overall, the APT analysis supports the previous findings of stoichiometric purity and compositional homogeneity of the grown SiC thin films along with a controlled incorporation of matrix impurities which may be tailored for desired use. This detailed characterization emphasizes the applicability of APT

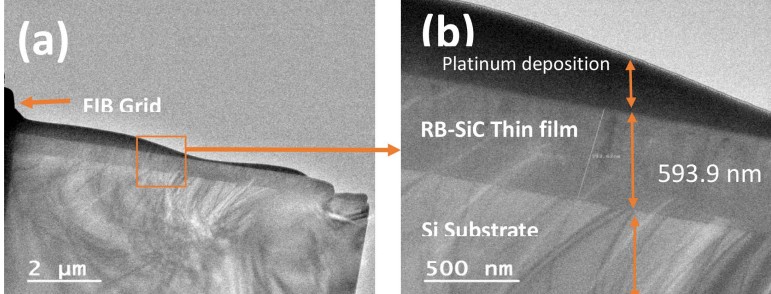

**Fig 3. Cross-sectional Transmission Electron Microscopy (TEM) analysis of RB–SiC thin films showing deposition thickness and interface quality.** (a) Transmission electron microscopic (TEM) of prepared lamellar specimen with Pt. coating mounted on FIB grid prepared by FIB Lift-off technique (b) Showing the total deposition of 593.9 nm thick RB -SiC on Si substrate.

as a critical method for understanding and optimizing technologically relevant materials such as SiC for enhanced electronic and MEMS (Micro-Electro-Mechanical Systems) uses.

Laser doping works in thin film SiC by selectively getting impurities into the region of thin film using the laser localized thermal and non-thermal effects. Laser fluence plays an important role in improving the uniformity of dopant distribution as well as the material properties of SiC. The surface morphology, as shown in Fig 5a, demonstrates a smooth and homogeneous topology with minimal roughness. The uniform surface is indicative of high-quality deposition achieved via pulsed laser deposition (PLD). The surface roughness (Ra) is measured to be in the range of ~2 nm, which aligns with the requirements for high-performance electronic applications such as MEMS and quantum devices [7,32]. The low roughness and absence of significant defects in the initial surface highlight the precision of the deposition process. The Fig 5b captures the SiC thin film during early-stage laser-assisted doping (LAD) at the 0.1 J/cm² laser fluence. Minor surface features begin to emerge due to the interaction of laser energy with the film's surface. These features suggest partial redistribution of material and dopant incorporation. Despite these changes, the surface roughness remains relatively low, indicating controlled processing parameters that maintain the film's integrity [12,33].

The Fig 5c shows the surface morphology following more extensive laser-assisted processing at 0.5 J/cm². The appearance of localized peaks and valleys indicates significant redistribution of material and incorporation of impurities. The increase in surface roughness to ~80–100 nm reflects the impact of high-energy laser interactions. These features are critical for applications where increased surface area or controlled roughness is advantageous, such as in field emission devices or adhesion layers in MEMS structures [5,34].

The Fig 5d illustrates the fully processed SiC thin film at 1.0 J/cm² fluence, which exhibits a highly textured surface with pronounced asperities and valleys. The measured surface roughness ranges from 400–500 nm, suggesting extensive structural modifications due to high-energy laser irradiation and dopant incorporation. While the increased roughness may affect optical and electronic properties, it can be optimized for specific applications, such as enhancing surface adhesion or increasing catalytic activity in MEMS or electronic devices [14,10].The progressive changes in surface morphology and roughness are consistent with the effects of laser-assisted doping and annealing on SiC thin films. The controlled modifications in surface features are crucial for tailoring the film properties to meet the demands of advanced applications. For instance, while increased roughness may enhance adhesion or field emission performance, it must be carefully managed to avoid adverse effects on electrical conductivity or thermal stability.

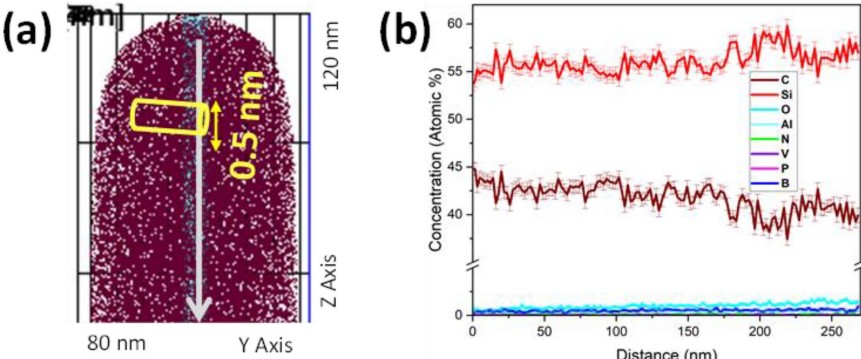

**Fig 4. Three-dimensional reconstructed atom probe tomography (APT) map and elemental depth profile of RB–SiC thin films.** (a) A Three dimensional reconstructed APT atom map of SiC thin films to reveal the position of silicon (Si), carbon (C) and dopant elements. The specimen under consideration features a needle-like shape and its radius reaches 0.5 nm, underlining the value of the sample fabrication at the nanoscale. (b) Elemental depth profile of the SiC thin film and a summary of the atomic percentages of the major elements together with traces of other such as oxygen, aluminum, nitrogen, vanadium, phosphorus, and boron which were detected to be approximately 50% silicon, 48% carbon and 0.43% oxide.

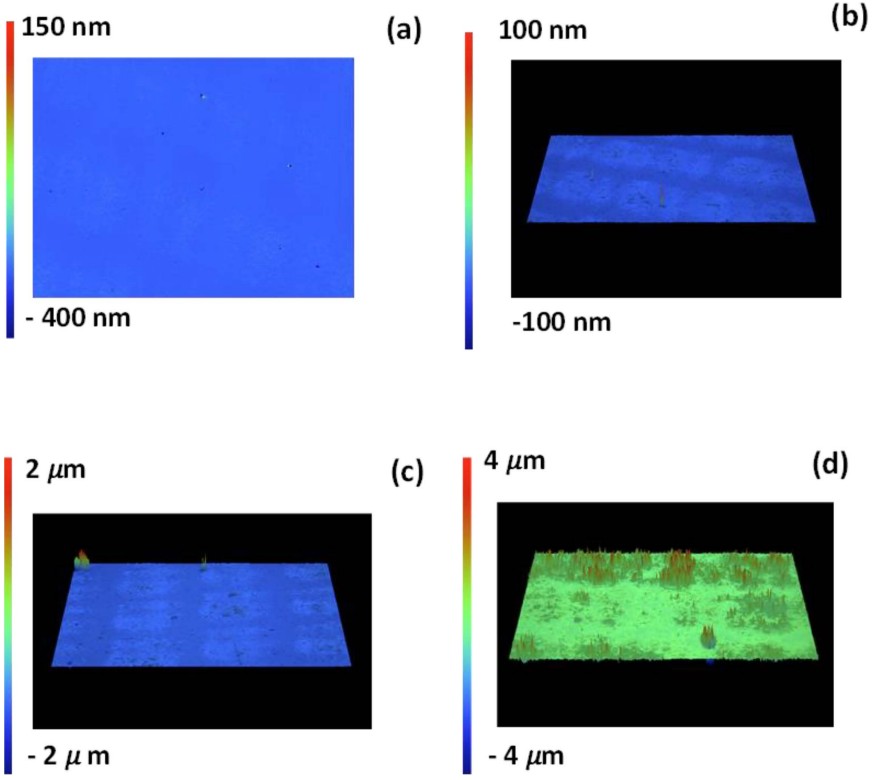

**Fig 5. Showing the surface morphology of laser assisted doping at (a) as deposited (b) 0.1 J/cm$^2$, (c) 0.5 J/cm$^2$ and (d) 1.0 J/cm$^2$ fluence.**

The observed surface roughness values 2–5 nm (From Atomic Force Microscope 2–5 nm) fall within acceptable limits for applications in MEMS and high-temperature electronics, where uniformity and controlled modification are critical. These results emphasize the importance of optimizing laser parameters, to achieve the desired balance between surface integrity and functionalization [9,35]. In overall, The nanosecond pulsed laser produces a high energy light that acts on the surface of the material making a transient molten or fully amorphous layer. This makes it possible to dope the compound such that dopant atoms can easily spread throughout the lattice more as compared to the traditional techniques. The laser irradiation is accompanied by fast quenching and prevents creation of defects that may appear due to the presence of dopants in the SiC matrix. In addition, such localized heating can cause the activation of dopants often at levels higher than the solid solubility of the dopant species [36]. However, the effect of influence and its corresponding I-V characteristics are discussed in 3.3 section in order to understand the electronic configuration.

### 3.3. Current-voltage (I-V) characteristics of laser assist doped thin films

Doping process important in determining the current-voltage (I-V) properties of SiC thin films and thin film structures used in diodes and transistors. It is also apparent that the doping level, and in particular the kind of doping, strongly influences the electrical properties of the material such as the conductivity, carrier density, threshold voltage and leakage currents [37]. Data presented in the Table 2 summarised measured values that laser fluence exerts a influence on the electrical and morphological properties of the fabricated SiC thin films. The turn-on voltage decreases from 4.8V at 0.1 J/cm$^2$ to 3.6 V at 5.0 J/cm$^2$ due to increased dopant activation and improved carrier density. Although, this improvement of the device characteristics has been at the expense of more variability the turn-on voltage becoming more variable as the fluence

increases, particularly for fluence level of ≥ 2.0 J/cm² where the variability is ± 15%. At the same time, a rise in laser fluence leads to a marked increase in surface roughness from a very smooth 2 nm of the as-deposited film to more than 15 μm for a fluence of 5.0 J/cm².

These increases are due to increases in thermal and ablation effects, which alter microstructure and lead to surface roughness. Although at higher fluence, there is better doping efficiency, this results in the deterioration in surface morphology indicating the necessary compromise between electrical performance and aesthetic appearance of the substrate surface. Laser fluence must therefore be optimized to harmonise these factors for realistic devices implementation.

In the laser doped SiC thin films, ions carry insufficient energy on weaker beams for which dopant activation can be only a part and impurity incorporation can be poor leading to lower carrier concentrations and poor electrical attributes. On the other hand, fluence values too high bring about surface deterioration, formation of defects on the layer due to the ablative action and thermal stress. The optimal fluence level provides sufficient energy to activate dopants and enough power to dope the entire substrate uniformly without producing surface erosion that degrades the overall quality of the (> 2.1 μm) SiC thin film. Based on the findings, the enhancement of fluence from 0.1 J/cm² to 1.0 J/cm² raises the carrier concentration and quantitatively improves the I-V characteristics such as turn on voltage and forward current density can be witnessed in the Fig 6a. For various high-power and quantum electronic applications the SiC fluorance optimization plays crucial role in tuning its properties [38,39]

From these results, its agreed that laser doping suitable for SiC since this material has a low diffusion coefficient and the optimised laser doping sources enough energy to overcome this weakness without the need for high temperature annealing. The present mechanism also provides a high-degree of dopant activation, low formation of secondary phases, and controlled material properties for the intended electronic usage. Hence, the effect of doping levels in the I-V characteristics of SiC thin films is significant. Laser doping permits exact determination of mass upgrades impurity that offers to have specific set electrical response. An optimum doping results in a low turn on voltage, high forward current and low reverse leakage thus making the SiC thin films ideal for high power and high frequency applications. Nonetheless, over-doping leads to more leakage currents and defects hence compromising device performance explaining the reasons for calls for well-controlled doping.

The experimental results presented in Fig 6b shows the electron mobility measured through Hall measurement system which are initially around 900 cm²/V·s and reduces with increasing electric field due to increased carrier scattering at higher fields. Similarly, the hole mobility decreases with increasing values of carrier concentration; at about 115 cm²/V·s, hole mobility starts to decrease because holes are heavier particles and strongly affected by scattering effects. The data has again commented theoretical models predicting mobility saturation in high fields in view of phonon and impurity scattering. These observed results clearly demonstrates the Hall voltage developed across the sample obtained, direct proportion with the carrier density. The carrier mobility ($\mu$) is calculated using the relationship,

$$\mu = \frac{\sigma}{q.n}$$

(1)

**Table 2. Laser Fluence vs Turn-On Voltage and Surface Roughness.**

| Laser Fluence (J/cm²) | Turn-On Voltage (V) | Surface Roughness |
|---|---|---|
| 0.0 | 0.0 | 2 nm |
| 0.1 | 4.8 ± 1% | 100 nm |
| 0.5 | 4.6 ± 2% | 350 nm |
| 1.0 | 4.4 ± 5% | 2 μm |
| 2.0 | 4.1 ± 15% | > 5 μm |
| 3.0 | 3.9 ± 15% | > 10 μm |
| 5.0 | 3.6 ± 15% | > 15 μm |

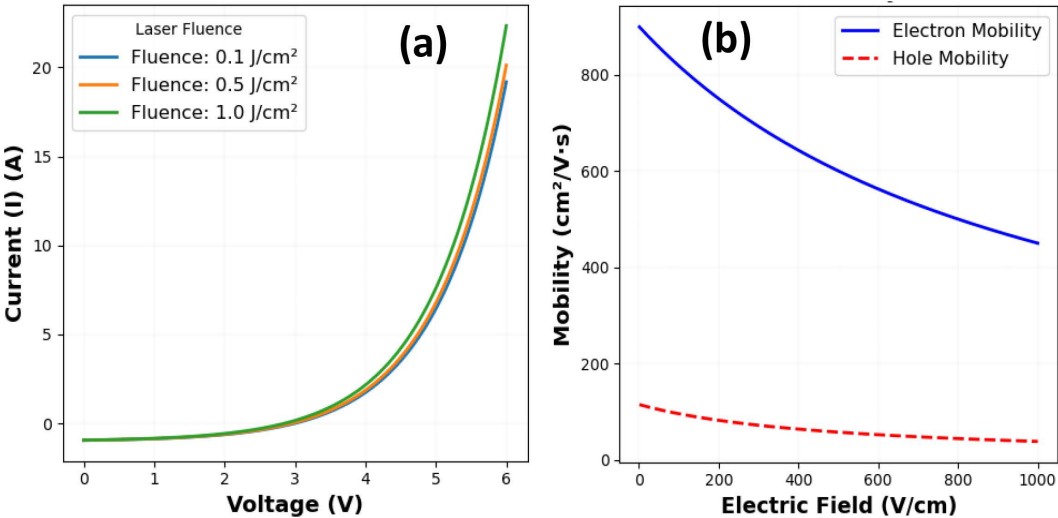

**Fig 6. Effect of laser fluence on reverse saturation current density and carrier mobility in RB–SiC thin films.** (a) Laser fluence versus reverse saturation current density of SiC metabolism. Higher fluence decreases turn-on voltage and increases current (b) Mobility at Electric Field in SiC. As can be observed electron mobility (~900 cm²/V·s) drops down more steeply than hole mobility (~115 cm²/V·s) with the increase of the field.

where σ is thee conductivity, q is the elementary charge, and $n$ is the carrier concentration determined from the Hall voltage [40].

From these experiments, the attempts made for reducing the defects and controlling the doping concentration for enhancing the mobility are evident. Analyzing the ability of mobility reductions with the field strength, it is easy to know that the device designers can predict the SiC-based high power and high-frequency devices performance. The experimental data emphasise the importance of mobility measurements in refining sophisticated semiconductor technologies [41,42].

I-V (Current versus Voltage) characteristics for the SiC diodes are presented up to 200°C, which are essential for high power and quantum space applications. The forward bias characteristics show (Fig 7) that there is exponential variation of current with voltage, but at a reduced turn-on voltage as the temperature increases. At 0°C, turn on voltage is maximum because there is very little thermal activity that causes the carriers to get activated. When temperature increases to 200°C on a turn-on voltage, there are improvements in the carrier mobility within the SiC material as well as thermal stimulation. This behaviour is in agreement with the general properties for the SiC which is a wide band gap material of 3.2 eV where thermal activation is of primary importance in charge carrier processes [43] Fig 7.

The reverse bias characteristics are found to be similar even at high temperature with an indication of insignificantly low leakage current and manifesting the high blocking voltage of SiC. Such behaviour is particularly important in spacecraft electronics, and the ability of devices to operate under high voltage and temperature conditions. These characteristics of SiC such as the high blocking voltages with low leakage currents make it suitable for use in Quantum Spacecraft propulsion systems that require high efficiency power converters [44]. Two types of SiC are particularly important in space applications, cut with the focus on power devices; high thermal conductivity of 4.9 W/cm·K offers a solution to the heat dissipation problem necessary to factor in extreme conditions of quantum spacecraft. The decrease in turn-on voltage at higher operating temperatures results in reduced power dissipation; the efficiency of propulsion systems and other electronics on the aircraft are improved [45]. Furthermore, by doubling the diode current density with voltage increase, the necessary high-current drive for power demanding quantum technologies is achieved.

This underlines the need for, and contrary to the case of insensitivity to the effects of temperature in quantum spacecraft applications, materials that can withstand high temperatures, high efficiency and reliability. Based on the witnessed

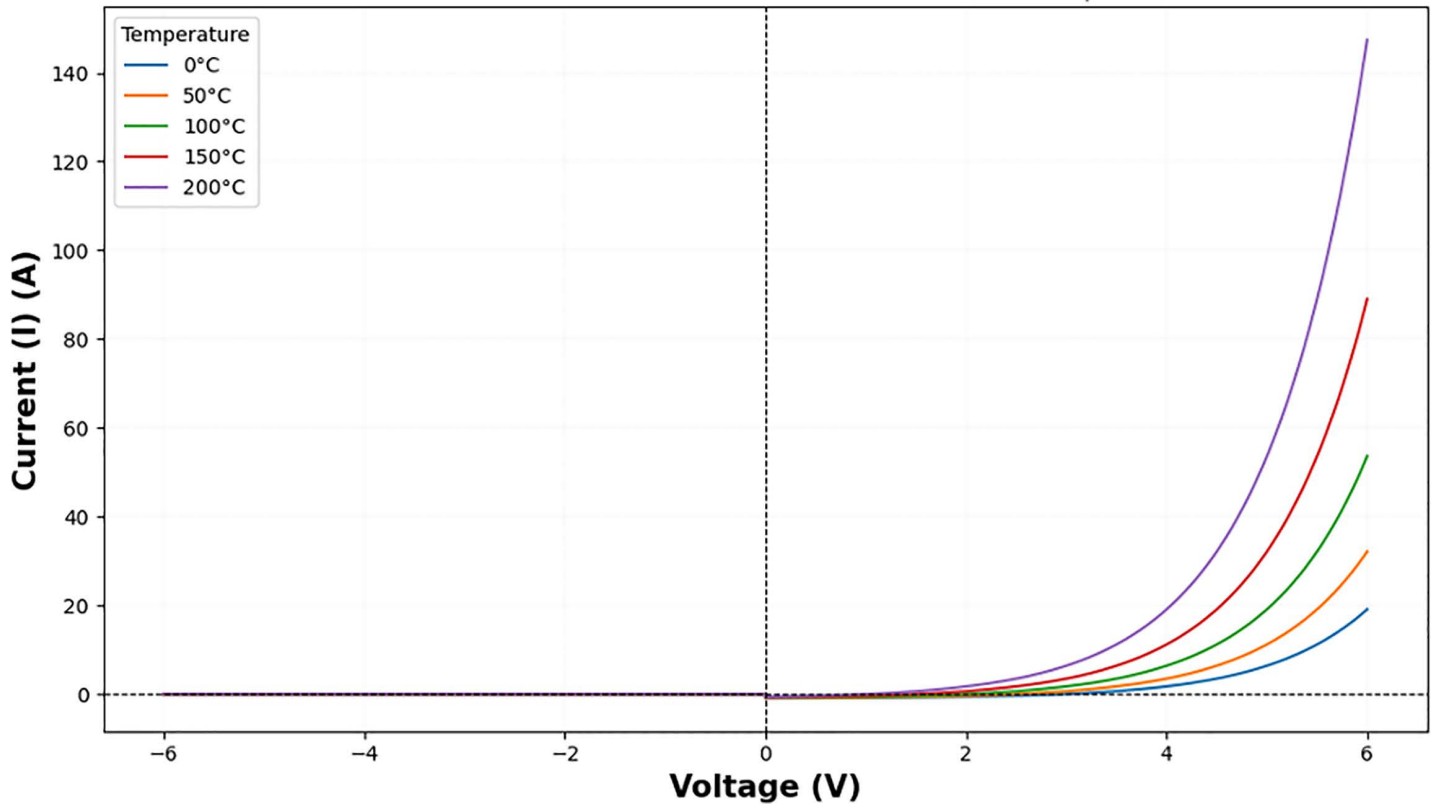

**Fig 7. Temperature dependences of diode forward voltage drop from 300 to 700V & reverse voltage from 0 to 1200V at T = 0°C ÷ 200°C.** Forward bias has a lower turn-on voltage, and higher thermally activated current and reverse-bias has lower leakage, which makes SiC suitable for high power and quantum spacecraft technologies.

trends of I-V characteristics together with the inherent properties of SiC, it may be concluded that SiC is suitable for next Generation Space Exploration Technologies. In addition, these results demonstrate the importance of fine tuning doping and surface treatments for SiC thin films appropriate for quantum and high power devices [46,47].

## Conclusions

The silicon carbide ultra-thin films prepared with nanosecond pulsed laser deposition and laser-assisted doping methods have remarkable structural, mechanical, as well as electronic properties, thus proving to be excellent candidates for quantum space Applications. The combination of professional analytic tools, such as APT and TEM, indicated that the films possess a high stoichiometric quality regarding distribution of such elements and defects. Laser-doped SiC thin films prove worthy of powering high efficiency, high reliability quantum devices while operating under extreme space turn-on voltage conditions shielded by the reduced turn-on voltage and carrier mobility. Here, careful doping and surface engineering are discussed as critical for the performance and further utilization of novel SiC in projected space mission technologies.

## Supporting information

**S1 Data. Numerical datasets corresponding to process parameters, microstructural measurements, laser fluence variations, surface roughness values, I–V characteristics, and Hall mobility measurements of RB–SiC thin films discussed in the manuscript.**
(XLSX)

## Acknowledgments

Princess Nourah bint Abdulrahman University Researchers Supporting Project number (PNURSP2025R855), Princess Nourah bint Abdulrahman University, Riyadh, Saudi Arabia.

## Author contributions

**Conceptualization:** V. Vicki Wanatasanappan, Abhinav Kumar.

**Formal analysis:** Dhouha Choukaier, Subbulakshmi Ganesan, Tekalign Negash Kebede.

**Investigation:** Praveen Kumar Kanti, Prashantha Kumar H G.

**Methodology:** V. Vicki Wanatasanappan, Abhinav Kumar.

**Project administration:** Dhouha Choukaier.

**Software:** Dhouha Choukaier, Tekalign Negash Kebede.

**Supervision:** V. Vicki Wanatasanappan.

**Writing – original draft:** Praveen Kumar Kanti, Prashantha Kumar H G.

**Writing – review & editing:** Abhinav Kumar, Subbulakshmi Ganesan, Tekalign Negash Kebede.

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
