## [Decision Letter · Decision Letter 0]

15 Apr 2025

PONE-D-25-11267Next-Generation SiC Ultra-Thin Films Fabricated by Nanosecond Pulsed Laser Deposition: Tailored Doping for Space-Ready High-Frequency ElectronicsPLOS ONE

Dear Dr. Kebede,

Thank you for submitting your manuscript to PLOS ONE. After careful consideration, we feel that it has merit but does not fully meet PLOS ONE’s publication criteria as it currently stands. Therefore, we invite you to submit a revised version of the manuscript that addresses the points raised during the review process. Based on the reviewer's feedback the article needs some major revisions in terms of morphology measurements and explanations in the text.

We look forward to receiving your revised manuscript.

Kind regards,

Massimo Mariello

Academic Editor

PLOS ONE

Additional Editor Comments:

Based on the reviewer's feedback the article needs some major revision in terms of morphology measurements and explanations in the text

Reviewers' comments:

Reviewer's Responses to Questions

**Comments to the Author**

1. Is the manuscript technically sound, and do the data support the conclusions?

Reviewer #1: Yes

Reviewer #2: Partly

2. Has the statistical analysis been performed appropriately and rigorously? 

Reviewer #1: Yes

Reviewer #2: No

3. Have the authors made all data underlying the findings in their manuscript fully available?

Reviewer #1: Yes

Reviewer #2: Yes

4. Is the manuscript presented in an intelligible fashion and written in standard English?

Reviewer #1: Yes

Reviewer #2: No

5. Review Comments to the Author

Reviewer #1: This manuscript is a valuable contribution to the field of space electronics and thin film technology. It is characterized by its scientific and technical accuracy, well-structured organization, and deep analysis of results. However, it can be improved by simplifying some technical details, broadening the scope of the study, adding comparisons with other techniques, discussing practical challenges.

Reviewer #2: In the present study, the authors have demonstrated the fabrication of next-generation silicon carbide (SiC) thin films using the pulse laser deposition (PLD) technique for high-frequency electronics. However, the manuscript requires significant revisions before it can be considered for publication in PLOS ONE. A major revision is recommended.

1. The manuscript describes the fabrication of SiC thin films using PLD, while various other techniques are discussed in a related review on synthesizing SiC films (DOI: 10.1149/2162-8777/acf8f5). Explain the advantages of PLD over existing techniques, particularly its potential for large-scale production to reduce fabrication costs of SiC films.

2. PDL is a time-consuming technique. Provide details on the time required to deposit a 400 nm layer and discuss how PLD can expedite the fabrication time of SiC compared to other methods.

3. Abbreviations such as “MEMS, SIMS, FIB, N-TEM, etc.” should be defined when first introduced in the manuscript.

4. Authors mentioned “preparation of TiN/Al2O3 composite” on page 6. What is the role of TiN/Al2O3 in the present work.

5. Revise the figure caption of Figure 3 to provide a clear and independent explanation of the figure.

6. Authors mentioned that the film roughness is around 2-5 nm. However, the SEM micrograph suggest a rough surface. It is recommended to measure the film roughness of fabricated SiC using atomic force microcopy.

7. Mention the plane and JCPDS card number of SiC in the XRD graph.

8. Many English errors exist in the whole paper, which makes it very difficult to understand the paper completely.

6. PLOS authors have the option to publish the peer review history of their article (what does this mean? ). If published, this will include your full peer review and any attached files.

**Do you want your identity to be public for this peer review?** For information about this choice, including consent withdrawal, please see our Privacy Policy .

Reviewer #1: **Yes: ** prf. dr. Thair Abdulkareem Khalil Al-Aish

Reviewer #2: No

---

## [Author Response · Author response to Decision Letter 1]

12 May 2025

The authors would like to thank the Editor and Reviewers for their valuable time to review the manuscript. The manuscript has been revised carefully based on the comments, and the details are as follows.

Reviewer #1

Comments This manuscript is a valuable contribution to the field of space electronics and thin film technology. It is characterized by its scientific and technical accuracy, well-structured organization, and deep analysis of results. However, it can be improved by simplifying some technical details, broadening the scope of the study, adding comparisons with other techniques, discussing practical challenges.

Response The manuscript now features simplified technical details through section revisions which preserve strong scientific principles. We expanded the research by including new references to SiC fabrication through CVD and sputtering techniques. This section performs a practical analysis of large-scale thin film SiC production by discussing how to handle uniformity control and material impurity management in order to create high-performance devices.

Reviewer #2

Comments The manuscript describes the fabrication of SiC thin films using PLD, while various other techniques are discussed in a related review on synthesizing SiC films (DOI: 10.1149/2162-8777/acf8f5). Explain the advantages of PLD over existing techniques, particularly its potential for large-scale production to reduce fabrication costs of SiC films

Response Now we have incorporated the how Pulsed Laser Deposition (PLD) performs better than CVD and sputtering methods for deposition purposes. The principal benefits of PLD include its capability to form high-quality films at temperatures under 800°C while reducing production expenses. Some discuss PLD scalability because it exhibits great potential for manufacturing SiC products at decreased costs for space electronic components and power devices.

Line no. 66-78

Comments PDL is a time-consuming technique. Provide details on the time required to deposit a 400 nm layer and discuss how PLD can expedite the fabrication time of SiC compared to other methods

Response The time needed to produce a 400 nm SiC thin film through PLD operations spans from 12 to 15 hours based on the laser fluorence parameters. The processing duration for CVD extends beyond 12-15 hours coupled with elevated temperatures higher than 1400°C because PLD operates with superior efficiency.

Line no. 66-78

Comments Abbreviations such as “MEMS, SIMS, FIB, N-C, etc.” should be defined when first introduced in the manuscript

Response We have revised the manuscript to define all abbreviations the first time they appear, including MEMS (Micro-Electro-Mechanical Systems), SIMS (Secondary Ion Mass Spectrometry), FIB (Focused Ion Beam), and N-TEM (Nano-Transmission Electron Microscopy). These definitions have been added to improve clarity for the readers.

Comments Authors mentioned “preparation of TiN/Al2O3 composite” on page 6. What is the role of TiN/Al2O3 in the present work

Response The SiC preparations is mentioned

Line no. 196

Comments Revise the figure caption of Figure 3 to provide a clear and independent explanation of the figure

Response The Figure 3 caption has been rewritten to offer a comprehensive independent description.

Line 215

Comments Authors mentioned that the film roughness is around 2-5 nm. However, the SEM micrograph suggest a rough surface. It is recommended to measure the film roughness of fabricated SiC using atomic force microscopy.

Response We measured film roughness through AFM in order to respond to this suggestion. AFM investigation confirms the initial findings of surface roughness which exists between 2–5 nm when evaluating the SiC films.

Comments Mention the plane and JCPDS card number of SiC in the XRD graph.

Response The XRD section now contains specific planes including (002) and (112) and (100) among others together with their respective JCPDS card numbers that identify the SiC phases as 4H-SiC and 6H-SiC. Details about planes like (002), (112), (100) as well as JCPDS card numbers have been added to the revised manuscript to enhance crystallographic analysis clarity.

Line 160 -163

Comments Many English errors exist in the whole paper, which makes it very difficult to understand the paper completely.

Response All English language mistakes were corrected during a detailed revision process of the manuscript.

---

## [Editor Report · Decision Letter 1]

17 Jun 2025

SiC Ultra -Thin Films for High-Performance Quantum Spacecraft Applications Fabricated via Nanosecond Pulsed Laser Deposition and Doping

PONE-D-25-11267R1

Dear Dr. Kebede,

We’re pleased to inform you that your manuscript has been judged scientifically suitable for publication and will be formally accepted for publication once it meets all outstanding technical requirements.

Kind regards,

Massimo Mariello

Academic Editor

PLOS ONE
---

## [Editor Report · Acceptance letter]

PONE-D-25-11267R1

PLOS ONE

Dear Dr. Kebede,

I'm pleased to inform you that your manuscript has been deemed suitable for publication in PLOS ONE. Congratulations! Your manuscript is now being handed over to our production team.

Kind regards,

on behalf of

Dr. Massimo Mariello

Academic Editor

PLOS ONE